# Smarter Evolution: Enhancing Evolutionary Black Box Fuzzing with Adaptive Models

**DOI:** 10.3390/s23187864

**Published:** 2023-09-13

**Authors:** Anne Borcherding, Martin Morawetz, Steffen Pfrang

**Affiliations:** 1Fraunhofer Institute of Optronics, System Technologies and Image Exploitation IOSB, 76131 Karlsruhe, Germany; steffen.pfrang@iosb.fraunhofer.de; 2KASTEL Security Research Labs, 76131 Karlsruhe, Germany; 3Karlsruhe Institute of Technology, 76131 Karlsruhe, Germany; morawetz.martin@gmail.com

**Keywords:** security testing, black box fuzzing, industrial control systems

## Abstract

Smart production ecosystems are a valuable target for attackers. In particular, due to the high level of connectivity introduced by Industry 4.0, attackers can potentially attack individual components of production systems from the outside. One approach to strengthening the security of industrial control systems is to perform black box security tests such as network fuzzing. These are applicable, even if no information on the internals of the control system is available. However, most security testing strategies assume a gray box setting, in which some information on the internals are available. We propose a new approach to bridge the gap between these gray box strategies and the real-world black box setting in the domain of industrial control systems. This approach involves training an adaptive machine learning model that approximates the information that is missing in a black box setting. We propose three different approaches for the model, combine them with an evolutionary testing approach, and perform an evaluation using a System under Test with known vulnerabilities. Our evaluation shows that the model is indeed able to learn valuable information about a previously unknown system, and that more vulnerabilities can be uncovered with our approach. The model-based approach using a Decision Tree was able to find a significantly higher number of vulnerabilities than the two baseline fuzzers.

## 1. Introduction

Attackers potentially attack individual components of production systems from the outside. This is, among other things, facilitated by the high connectivity introduced by Industry 4.0. Individual control systems under attack are, for example, SCADA systems and Industrial Control Systems (ICSs) such as Programmable Logic Controllers (PLCs) [1]. To strengthen the security of those components following an in-depth defense strategy, several layers of security should be implemented. One of these layers necessary to improve the security of smart production systems is to ensure the robustness and resilience of the used components by performing thorough security tests. Part of these thorough tests is to perform black box tests against the network interfaces of the components. This is especially required by IEC 62443, the standard for security in automation and control systems [2]. Black box tests do not presume any information on the internals of the System under Test (SUT). Hence, black box tests are also necessary if the SUT uses components from third-party vendors from which no information of the internals can be accessed.

The topic we focus on in this work is black box *fuzzing*. Fuzzing is a testing technique that first generates random input. This input is sent to the SUT, and the fuzzer monitors how the SUT reacts to this input. In order to give the randomly generated test cases more structure, the fuzzer needs some kind of guidance. For this guidance, several approaches have been proposed. One approach is *coverage-guided* fuzzing, in which the fuzzer uses the new code coverage from a test case triggered as guidance for the progression of the test (e.g., [3,4]). However, a requirement for coverage-guided fuzzing is access to the code coverage. In a black box test, this information is not accessible. A different approach for guided testing is *evolutionary* testing. For this, the fuzzer generates new test cases based on an Evolutionary Algorithm (EA) and thus takes the effectiveness of the previous test cases into account while generating the new test cases. As a basis, a measure for the effectiveness of the test cases is needed. In gray box tests, the code coverage is often used as a measure [5]. However, as mentioned before, this information is unavailable in black box tests.

In this work, we propose a means to approximate the missing information in a black box test in order to provide guidance to a fuzzer. To this end, we aim to automatically learn information on the internals of the SUT by leveraging the little information that *is* available in a black box test. This available information includes, for example, whether the SUT crashed as a response to a given test case. We use the newly learned information on the internals of the SUT to guide a fuzzer based on an EA.

In more detail, we approximate the information necessary for the EA by using machine learning models. The models provide approximated information on the internals of the SUT, which is then used in several steps of the EA. The population the EA works with consists of the test cases to be tested against the SUT. Then, the EA needs to solve two challenges: (I) mutating existing test cases to produce a new offspring and (II) ranking the test cases in the offspring to generate a new population. We propose using the trained models for both of these challenges. As a basis, these models adaptively learn to approximate the behavior of the SUT and are thus able to make a prediction of how the SUT will behave with respect to a given test case. Based on this knowledge, the models can guide the mutations in the direction of a potential crash of the SUT as well as rank new test cases based on the probability to crash the SUT. With this, we propose a combination of evolutionary fuzzing and automatically trained models. The models help with the test case selection and mutation in a short-term manner, and the evolutionary principles provide long-term improvements and a feedback loop. Our approach aims to find vulnerabilities in ICS early in the development life-cycle to avoid these vulnerabilities being exploited by attackers. With this, we aim to enable manufacturers and researchers to find vulnerabilities and close them either directly or by using a responsible disclosure strategy.

### 1.1. Contributions

We present an approach to performing evolutionary gray box fuzzing in a black box setting by leveraging machine learning models that approximate the information which is unavailable in a black box test. This allows for guided fuzzing in a black box setting, which is needed for a thorough test of components, especially in smart production ecosystems. In particular, this allows for thorough tests, even if information on internal parts of the components is not accessible. We call our approach Smevolution and implement it based on the security testing framework ISuTest^®^ [6]. We evaluate Smevolution by running different fuzzer configurations against a SUT with artificial vulnerabilities. These artificial and known vulnerabilities allow for a deep analysis of the behavior of the fuzzers. In addition to the general performance, we analyze how different levels of detail with respect to the feedback the fuzzer receives influence the performance of the fuzzer.

In summary, our work makes the following two main contributions:Smevolution allows gray box test approaches to be used in a black box setting by approximating missing information using automatically trained models.We analyze what impact the choice for the level of detail the fuzzer receives as feedback has on the fuzzers performance.

### 1.2. Related Work

Our approach is to automatically train a model of the SUT, which will then be used to guide and support an evolutionary fuzzing approach with respect to the mutation and the fitness evaluation. To the best of our knowledge, our work is the first that chooses this approach to improve evolutionary fuzzing. Nevertheless, related publications in the domains of evolutionary fuzzing, fuzzing based on machine learning, and automated learning of models exist and are presented in the following paragraphs.

**Evolutionary Fuzzing.** There are several approaches to evolutionary fuzzing, both in black box and gray box settings. Prominent representatives of gray box evolutionary fuzzing are AFLNet [7], which is a network fuzzer based on AFL, and HongFuzz [8], a fuzzer implemented by Google. Both fuzzers rely on an instrumentation of the SUT in order to gather the data necessary for the fuzzing process, such as code coverage. In addition, AFLNet analyzes the SUT’s response codes to identify the current state of the SUT. Furthermore, several works have been presented in the domain of black box evolutionary fuzzing. Appelt et al. [9] present an approach to fuzz web application firewalls.Based on the paper’s title, we will call their approach MLDriven in the following. MLDriven uses a Decision Tree (DT) to generate SQL injection test cases, which have the requirement to bypass the firewall first. Specifically, this DT is used to identify the parts of a test case that are responsible for a firewall bypass. These parts will not be changed to retain the bypass functionality of the test case. MLDriven only uses binary feedback to train the DT, namely the binary information regarding whether the test case was able to bypass the firewall. We will use the approach of MLDriven as one of the approaches used for Smevolution (see Section 2.2.4). Duchene et al. [10] present a different approach to web application firewall fuzzing, and also use an evolutionary approach.In contrast to MLDriven, the authors do not use a DT for the fitness evaluation but use the response DOM as well as taint inference to evaluate the fitness of a test case. With this, they do not learn an explicit model of the behavior of the SUT. Chen et al. [11] present an evolutionary fuzzing framework for cyber-physical systems based on machine learning models.The framework uses the sensor values of the tested cyber-physical system as an input for the machine learning models. Unlike in network fuzzing, the sensors in cyber-physical systems provide continuous data, which can be used by the machine learning models in a more direct way to evaluate and manipulate the test cases. Note that none of the approaches is concerned with network fuzzing.

**Machine Learning for Fuzzing.** In the literature, machine learning approaches have been used at various stages of the fuzzing process. For example, machine learning has been used for seed selection [12], input generation [13], and for input selection [14]. Wang et al. [15] provide a detailed overview and classification of these approaches.Our approach is to combine an evolutionary algorithm with automatically and continuously trained models for (I) input generation and (II) seed selection of test cases. To the best of our knowledge, there is no such approach in the literature.

**Automated Learning of Models.** For general automata learning, various algorithms and approaches have been proposed [16]. In addition, some approaches have been published with regard to automated learning of models for fuzzing. Doupé et al. [17] propose an approach for automated black box state machine inference for web applications.This state machine is then used to guide the fuzzing process. Based on the work by Doupé et al. [17], Borcherding. et al. [18] present a generic framework for state machine inference in black box testing of web applications.Gascon et al. [19] train a Markov Chain representing the states of a network protocol and use this knowledge to guide the fuzzing process.McMahon Stone et al. [20] automatically learn a protocol state machine in a gray box setting.However, these approaches learn an explicit model of the SUT, which is then used to explicitly guide the fuzzing process. In contrast, we aim to learn an implicit representation of the SUT’s behavior. We will use this information to guide the mutations and to evaluate the fitness of test cases.

### 1.3. Background

Our work is located in the research domain of fuzzing, and we are using EAs and approaches from Machine Learning (ML). The following paragraphs will provide the background on EAs and machine learning necessary for the remainder of this work. In addition, we will give an overview of the security testing framework ISuTest^®^ and the SUT that we base our implementation and evaluation on, respectively.

#### 1.3.1. Evolutionary Algorithms

The term *Evolutionary Algorithm* comprises several search algorithms that incorporate parts of natural evolution [21]. One of the basic versions of an EA is shown in Figure 1 and explained below. Usually, an EA includes two sets of individuals: the population and the offspring. Individuals in the population are mutated or re-combined to form the new offspring. Based on a fitness function, a subset of individuals from the new offspring is selected to form the new population. Depending on how the mutation and the selection take place, the algorithm can solve a specific optimization problem. In our work, we aim to find the test case that leads to a crash of as many services of the SUT as possible. To follow the remainder of this work, a basic understanding of EAs is sufficient. Refer to the work of Bartz-Beielstein et al. [21] for a more detailed description of EAs.

#### 1.3.2. Machine Learning Approaches

*Machine Learning* is a domain that spreads in nearly all fields of research, and many resources regarding the fundamentals of ML exist (e.g., [22]). Thus, we are restricting this section to the information necessary to follow the remainder of this paper. This section also broadly describes how we are using the ML models for our work. We give more details on our approach in Section 2.2.

**Neural Networks.** Neural Networks (NNs) consist of a network of single nodes, inspired by the inner workings of the human brain. Based on their construction, NNs approximate complex problems by a high dimensional but smooth function, which is differentiable. We are using NNs to create a differentiable representation of the SUT’s behavior. In more detail, this behavior will be represented by a mapping from a test case to the services of the SUT that crash in reaction to this test case. Then, we can calculate the gradient of this function in order to guide our tests in the direction of a value that leads to more crashing services.

**Decision Trees.** DTs use a tree-based structure to classify a given data point into one of several classes. Each node of the tree corresponds to one decision that is taken to come to the final conclusion, and each leaf corresponds to one class. To classify one data point, the tree is walked based on the outcomes of the decisions on the inner nodes. The final classification decision of the DT depends on the leaf node we end up with. We use a DT to represent which services will fail as a reaction to a given test case. Since a DT explicitly stores the decisions it takes to classify a test case, we can identify the properties of a test case that need to be changed in order to get it classified as a test case that leads to more crashing services. We use this property to guide our tests.

**Support Vector Machines.** Support Vector Machines (SVMs) can be used for classification and for regression tasks. They are designed to split a set of data points with the goal to maximize the distance between the resulting clusters. For our work, we are using an SVM for the classification of the test case, similar to the approaches based on an NN and on a DT presented above. In contrast to the previous two models, we only use the SVM for classification, not for mutation guidance.

#### 1.3.3. ISuTest^®^

ISuTest^®^ is a framework for security tests against networked devices, which focuses on black box fuzzing via Ethernet [6]. It enables users to detect formerly unknown vulnerabilities in ICs. ISuTest^®^ is divided into three main components: *Environment*, *ISuVentory*, and *ISuGUI*. The Environment runs tests against the SUT and monitors the SUT’s behavior. Multiple Environments can be connected to one central ISuVentory. The ISuVentory provides test jobs for SUTs, which are then pulled and executed by the respective Environments. After the execution, Environments push their test results, network packet captures, and log files to the ISuVentory. The user interacts with ISuTest^®^ using the graphical user interface ISuGUI. Extensions of ISuTest^®^ can be employed as ISuVentory plugins. In particular, plugins can create and evaluate test jobs, including their results in an automated manner.

#### 1.3.4. VulnDuT

The VulnDuT is an intentionally vulnerable Device under Test and is part of ISuTest^®^. The VulnDuT is set up as a docker container with two network interfaces: the testing interface and the configuration interface. On the testing interface, several services that run in the container can be accessed, e.g., the IP stack or an HTTP server. Additionally, the VulnDuT service can be contacted by using the VulnDuT protocol. On the configuration interface, several predefined scenarios can be enabled. Each scenario defines which services will crash if a special input is sent to the VulnDuT using the VulnDuT protocol. The input can be contained in one single network packet as well as in a series of network packets. As such, a scenario represents a deterministically occurring vulnerability. A single-packet example for a scenario to crash is if an Integer with a value of 64 is received. A multi-packet example for a scenario to crash is if the sum of received Integer values is larger than $2^{32}$. We will define several new scenarios for the VulnDuT to evaluate Smevolution (see Section 2.4.1).

## 2. Materials and Methods

In this section, we first formulate our research questions. Then, we describe our approach and how we implement it based on ISuTest^®^ (Fraunhofer IOSB, Karlsruhe, Germany). In addition, we present our evaluation setting, including information on how we aim to answer the previously formulated research questions. We will publish the data generated during our experiments as well as code to produce the figures used in this paper upon publication of this work.

### 2.1. Research Questions

Our work aims to allow for the improvement of black box testing approaches by leveraging the little information that is available. We aim to achieve this by approximating the behavior of the SUT and by using it to enable the usage of a gray box evolutionary testing approach, even though the original setting is black box. Our work is driven by the following three research questions. Note that our evaluation focuses on the performance of fuzzers based on the different models and not on the models themselves. The ultimate goal is to find vulnerabilities as fast and reliably as possible, not to train models that are as accurate as possible.

***RQ1*** 
*Can automatically learned models represent meaningful information?*


The first research question is concerned with the basic question of whether the approach of automatically learned models has the potential to improve the fuzzing process. We evaluate this question by comparing the performance of fuzzers based on the learned models with the performance of two baseline fuzzers. As baseline, we use (I) a fuzzer that uses random decisions instead of questioning a model and (II) a fuzzer that uses the raw information on the number of crashed services as a selection criterion.

***RQ2*** 
*How much does the used information influence the performance of fuzzing?*


The second research questions is concerned with the specifics of the data used to train the models. We analyze how different levels of granularity of the feedback the fuzzer receives influence the performance of the model-based fuzzers. This also aims to answer the underlying question of how much information one needs to achieve better fuzzing results.

***RQ3*** 
*How much overhead do the models introduce?*


The models introduce an overhead if compared to a fuzzer that bases its fuzzing on random decisions. However, one needs to find a balance between the additional overhead and the potentially improved fuzzing. We aim to answer this question by measuring the number of test cases the model-based fuzzers can achieve in a given time frame in comparison to a fuzzer based on random decisions. Note that the models do not require an explicit training phase before the fuzzing can start since the models are trained on the fly.

### 2.2. Approach

Our goal is to enable the use of evolutionary gray box fuzzing approaches in a black box setting. Thus, we present Smevolution, which approximates the missing information by automatically and adaptively learned models that represent the behavior of the SUT. The base of Smevolution is a basic EA, which is enriched and supported by the information represented by the models.

In the following, we first explain the structure of the EA and then how the automatically learned models support it. We will refer to the automatically learned models as *models* in the following.

#### 2.2.1. Evolutionary Algorithm

As a base for Smevolution, we use a basic EA, as shown in Figure 2. The algorithm includes a population and an offspring. In each round, new individuals are created by mutating individuals from the population, forming the new offspring. From this new offspring, the fittest individuals are selected based on a fitness function.

**Individuals.** The population and the offspring contain test cases for the SUT. In our case of network fuzzing, these test cases contain values for the fields of the network packet that is currently fuzzed. For example, assume that we want to fuzz two fields of a network packet from which one contains an unsigned Integer and one contains a String. In this case, a test case could look as follows:(1)[412,“AAAA”]

**Mutation.** New individuals are created by mutating individuals from the population. The goal of the mutation is to generate new test cases that will trigger new anomalies or vulnerabilities in the SUT. In the general black box setting, we do not have any specific information on how to mutate an individual in order to create a new individual that has a high probability of triggering an anomaly or vulnerability. Thus, mutations in a black box setting are usually done at random. In general, mutations for fuzzing usually consist of, for example, replacing or deleting parts of the test case or by adding new parts [5]. A mutated test case, which has been created by a bit flip in the last character, could look like the following:(2)[412,“AAAC”]

**Execution.** The test cases in the offspring are executed against the SUT and the behavior of the SUT in reaction to the test case is observed. These observations can then be used to determine the fitness of the test cases. The test cases are either executed individually to obtain detailed information on the behavior of the SUT in reaction to each single test case, or executed in groups to speed up the testing process. By executing the test cases, we get to know a label for each test case or test case group in the offspring regarding the corresponding behavior of the SUT. For example, assume that the SUT we executed our example test case against has three services, e.g., SNMP, TCP, and HTTP. If the example testcase from Equation (Equation 2) crashed the SNMP service, the resulting labeled test case could look like the following, indicating that the first service crashed. The label is represented as a bitstring, one bit representing one of the services of the SUT.
(3)[[412,“AAAC”],b‘100’]

**Selection.** From the newly created offspring, the individuals that have the highest fitness are selected to form the new population (survival selection). In our case, the fitness function used for this decision needs to determine how much a test case will contribute to finding new anomalies or vulnerabilities in the future.

#### 2.2.2. Automatically Learned Models

To enrich the previously presented EA, we learn and use a representation of the behavior of the SUT (see Figure 3). This representation helps to determine the fitness as well as to guide the mutation, and it is trained using the labeled offspring data (such as those presented in Equation (Equation 3)). Note that we train the models parallel to the fuzzing process and thus do not need a training phase at the beginning of the fuzzing.

In general, the models learn a representation of how likely it is that a test case will cause a crash in the SUT. Formally, the models approximate the following function:(4)$$f:H^{n}\to {\left\{0,1\right\}}^{m},$$
where *H* denotes the set of all possible hexadecimal values, *n* the length of a test case, and *m* the number of services of the SUT that are monitored during the test. Thus, the learned function represents a mapping from a test case in a hexadecimal representation to a bitstring representing the crashed services.

This function and the internal representation is then used as a fitness function for the test case in the context of the previously seen test cases, and, more importantly, as an indicator on how test cases should be mutated in the future to generate more promising test cases. How this indicator looks like in detail highly depends on the type of the learned model. In the following, we present three different instantiations of models that we designed and implemented for Smevolution and use in our evaluation. We present approaches based on a NN (A_NN), on a DT (A_DT), and on a SVM (A_SVM).

#### 2.2.3. Neural Network (A_NN)

Our first approach to model the behavior of the SUT is inspired by the work by She et al. [23]. She et al. [23] propose an approach to approximate a program’s branching behavior by a smooth function using NN. We build upon this idea and use an NNs to smoothly approximate a program’s crashes. As has been described for the general approach in Section 2.2.2, the model learns a continuous and differentiable function that maps a test case to the services that have been crashed by this test case. This function is then used to decide in which direction a test case needs to be mutated in order to find a test case for which more services crash (gradient ascent). Thus, A_NN follows a depth search strategy to find deeper vulnerabilities.

#### 2.2.4. Decision Tree (A_DT)

The second approach is inspired by the work by Appelt et al. [9]. As has been described in Section 1.2, the authors present an approach to use a DT for test case generation to test Web Application Firewalls. The DT is used to decide at which position of a test case a mutation should be performed. In more detail, the authors analyze the DT to find out which parts of the test case should be kept for future individuals and which should be mutated. In their setting, the parts of the test case that should be kept are the parts that help the test case to evade the Web Application Firewall.

We adapt the approach by Appelt et al. [9] and use a DT as follows. The DT learns a function that maps a test case to the services that fail during the execution of this test case. If the DT decides that a test case leads to the failure of some services in the SUT, we can automatically analyze which parts of the test case lead to this decision. This is possible because of the structure of a DT (see Section 1.3.2). Thus, these parts of the test case are marked to be not changed during a mutation to keep this property of the test case. The goal is to find a test case that leads to the failure of even more services. As a result, A_DT takes a depth search approach for the vulnerabilities and crashes.

#### 2.2.5. Support Vector Machine (A_SVM)

To analyze the impact of the enhancement of the mutation, we use a model that only enhances the fitness function as a third approach. For this model, we choose an SVM, roughly following the approach by Chen et al. [11]. Since this approach does not guide the mutation in a specific direction, we expect the approach to perform a broader search than the two approaches presented above.

### 2.3. Implementation

We implement Smevolution with a modular approach. Both parts of the implementation, the EA and the used model-based algorithm, are implemented as modules that are interchangeable. The EA is implemented as a wrapper that calls the model-based algorithm and manages the used datasets, such as the Population and the Offspring. The model-based algorithm implements the functions to mutate test cases and to support the EA with the test case selection. For our evaluation, we implement the aforementioned model-based algorithms (Section 2.2.3, Section 2.2.4 and Section 2.2.5). Nevertheless, our modular approach allows for an easy extension with new algorithms. We implement the aforementioned algorithms based on existing implementations of the underlying ML algorithms. We used the DecisionTreeClassifier provided by scikit-learn v1.1.1 for the implementation of A_DT, tensorflow v2.9.1 to build an NN as described by She et al. [23] for A_NN, and the SVM implementation by scikit-learn v1.1.1 for A_SVM. With this, we implement three model-based fuzzers, each based on one of the three presented algorithms. In addition, we implement two baseline fuzzers for our evaluation (see Section 2.4.2 for details).

Each of the algorithms has some hyperparameters that need to be set. We describe our choices of hyperparameters in the following. If not explicitly stated, we use the default parameters given by the underlying framework providing the implementation of the ML algorithm. For A_DT, we performed some preliminary experiments to decide on the maximum tree depth and the maximum number of leaf nodes. Since these values showed the most promising results, we choose a maximum tree depth of 8 and a maximum number of leaf nodes of 40. The NN used by A_NN uses the same architecture and parameters as used by She et al. [23]. In short, the NN has three fully connected layers and uses ReLU as the activation function for the hidden layers and sigmoid as the activation function for the output layer. For A_SVM, we used the OneVsRestClassifier provided by sklearn v1.1.1 using Support Vector Classification, with a linear kernel as a basis. To provide even more detail on the models, we publish the source code that we used to define the models (https://github.com/anneborcherding/Smarter-Evolution). Future work could include a more detailed analysis on how the choice of hyperparameters influences the final results of the fuzzing and whether the hyperparameters should be adapted to the SUT.

The interoperability of the algorithm and the EA is ensured through the usage of an interface. The interface provides the function signatures fit_encoder, train_estimator, rank_testcases, and generate_offspring. Each of those signatures have to be implemented for an algorithm to allow for its integration into the EA. Due to the differences in nature, each algorithm also has to come with its own encoding for test cases.

Once the EA and an algorithm have been combined, the execution conducts several steps, which can be seen in the flow chart in Figure 4. One round of the shown loop equals one generational iteration of the EA. After the initialization, each evolutionary round includes the following steps. First, the new offspring is generated based on the previous population and the mutators. Depending on the used algorithm, these mutations are influenced by the learned model. Then, the offspring is executed against the target and the results are used to update the archive. This archive includes all test cases that have been run against the target as well as information on how the target reacted to each test case. The newly gained information is then used to re-train the model, and the new model is used to rank the test cases and to gather the information for future mutations. In the following selection step, the new population is created. After this, a termination criterion is evaluated to decide whether the next evolutionary round will be started or whether the fuzzing process should be finalized. In our implementation, we include a time-based and a round-based termination criterion. For our evaluation (see Section 2.4), we used the time-based criterion and allowed a time budget of 24 h for each run.

Smevolution is integrated into ISuTest^®^ v2.1.0 as an ISuVentory Plugin (see Section 1.3.3). This allows us to easily evaluate its performance against the VulnDuT. ISuTest^®^ also provides an interface to run mboxSmevolution on real-world appliances. With this, our work can be extended to verify the results on real-world ICs (see Section 5).

### 2.4. Evaluation

In order to answer our research questions (see Section 2.1), we conduct several experiments. This section details our evaluation strategy and our experimental setup, while Section 3 presents the results of the experiments. We discuss these results in Section 4.

Our evaluation is based on the artificial SUT, VulnDuT, to analyze the algorithms’ behavior in detail. For our general evaluation setup, we follow the guidelines for fuzzing evaluations presented by Klees et al. [24]. Specifically, we run each configuration of the model-based fuzzers for 24 h 10 times to account for their randomness. Our evaluation is based on docker images, which are run on virtual machines based on Ubuntu 20.04.6 LTS with an Intel^®^ Xeon^®^ Silver 4216 CPU @ 2.10 GHz (12 cores) and 32 GB of RAM.

In the following, we present the vulnerabilities that we implement for the VulnDuT (Section 2.4.1), give more details on the fuzzers that we use as a baseline (Section 2.4.2), and then describe how we aim to answer the research questions (Section 2.4.3, Section 2.4.5, and Section 2.4.6), which we presented in Section 2.1. Table 1 shows the configurations of the fuzzers and the environment that we used during the evaluation. It will be detailed in the following sections.

#### 2.4.1. Vulnerabilities

We implement several artificial vulnerabilities for the VulnDuT in order to measure the vulnerability detection performance of the various model-based fuzzers and the baseline fuzzers. To this end, we use the artificial network protocol provided by VulnDuT. The network protocol is built upon UDP and includes three fields: an unsigned Integer, a signed Integer, and a String with a fixed length. Based on this network protocol, we define several vulnerabilities that will lead to crashes of different services of the VulnDuT. Table 2 shows the vulnerabilities and the services that are dropped if the corresponding vulnerability is triggered.

We implement the following types of vulnerabilities, with which we can analyze the behavior of the algorithms in different situations.

Independent vulnerabilitiesLinked vulnerabilities

The first group of vulnerabilities, the independent vulnerabilities, are not connected in any way. Finding these vulnerabilities can be compared to a *breadth-first search*. For example, these vulnerabilities include (I) a crash of HTTP triggered by an unsigned Integer value of $2^{32}-1$ (see Table 2a), (II) a crash of ICMP triggered by a signed Integer value of $-2^{31}$ (see Table 2b), and (III) a crash of SNMP triggered by a String starting with “KL” (see Table 2d).

In contrast, the linked vulnerabilities are vulnerabilities that are fairly close to each other. Closeness in this case corresponds to the number of mutations that need to be conducted to move from one vulnerability to the other. Finding these vulnerabilities is comparable to a *depth-first search*. The linked vulnerabilities are present in the String field of the used artificial protocol. We differentiate between three approaches of linked vulnerabilities. First, we implement linked vulnerabilities based on String matching that lead to more and more crashes (see Table 2c). Second, we implement similar vulnerabilities, but these only result in single different services crashing (see Table 2d). Third, we implement vulnerabilities that are triggered by ranges of signed and unsigned Integer values. These crashes lead to the same set of services crashing with every hit of the range (see Table 2a,b).

#### 2.4.2. Baseline Fuzzers

As a baseline for our evaluation, we use two different baseline fuzzers. On the one hand, we use a random fuzzer that performs the selection at random and uses the default mutation strategy provided by ISuTest^®^. This mutation strategy is based on fixed heuristics crafted for black box network fuzzing [6]. We will refer to this random fuzzer as A_RANDOM in the following. On the other hand, we use a baseline fuzzer that performs the selection directly based on the number of services a test case crashed (called A_BASELINE in the following). The model-based fuzzers use this information indirectly via the learned model. A_BASELINE also uses the default mutation strategy by ISuTest^®^.

Note that the goal of our work is to analyze whether the models are able to learn meaningful information and use such fuzzers that are less sophisticated as a baseline. Nevertheless, these baseline fuzzers are based upon mutation heuristics especially developed for the use in black box network fuzzing for ICS [6]. For future work, it would be interesting to compare Smevolution to additional state-of-the-art black box fuzzers for network protocols to analyze their general performance.

#### 2.4.3. RQ1—Model Impact

The first research question is concerned with whether the automatically learned models can represent meaningful information. We evaluate this by comparing the fuzzers based on the three algorithms (A_DT, A_SVM, A_NN) with the two baseline algorithms presented above. Through a comparison of the model-based fuzzers to the random fuzzer (A_RANDOM) and the baseline fuzzer (A_BASELINE), we can analyze how the used models influence the fuzzing performance. Furthermore, we analyze how the different fuzzers perform regarding the different vulnerability types presented above.

Based on our approach, we would expect that the model-based algorithms perform better than A_BASELINE and A_RANDOM, and that A_BASELINE performs better than A_RANDOM in terms of triggered vulnerabilities. In addition, we would expect that A_DT and A_NN perform better in finding linked vulnerabilities since A_NN performs more locality based mutation decisions and A_DT explicitly optimizes the number of crashes to be triggered. We expect this to be in favour of finding the linked String vulnerabilities in Table 2c.

#### 2.4.4. RQ2—Information Granularity

The second research question is concerned with how much the information granularity influences the performance of the fuzzers. For this, we analyze the impact of two different parameters for the feedback: the feedback dimension and the feedback interval.

For the feedback dimension, we introduce two different configurations. The first configuration includes full information on which services crashed during the test (*multidimensional information*). In more detail, the fuzzer receives the full byte string with full information, as presented in Equation (Equation 3). The second configuration reduces the information content to one bit (*unidimensional information*). This bit is set to 1 if at least one service has been crashed. We would expect that the model-based mutating strategies with the full information available find more vulnerabilities that lead to several crashing services. This expectation is based on the design of the mutation strategies, which aim to maximize the number of services that are crashed by a test case. Furthermore, the models might be able to learn more meaningful information if they have more detailed knowledge on the behavior of the SUT.

For the feedback interval, we choose three different intervals in which the fuzzer will receive feedback on the results of the previous test cases. In realistic scenarios of black box ICS fuzzing, a fuzzer will not receive feedback after each test case since this will introduce too much overhead. For each feedback cycle, the fuzzer needs to check for each of the monitored services whether this service still works. For example, the fuzzer needs to send an ICMP echo request and wait for the response to check whether ICMP is up and running on the SUT. To take this into account and to analyze the impact, we introduce different configurations for the feedback interval. The feedback interval represents the number of test cases that are sent to the SUT in a bulk before the fuzzer receives feedback. In our case, we set the number of test cases to be sent in bulk to 10, 50, and 100. Note that the fuzzer receives accumulated feedback on how the state of the services is after all of those test case have been sent to the SUT. As a result, the data the fuzzer receives becomes more inaccurate the larger the feedback interval is. After a feedback cycle, the SUT is reset in order to be in a known state for the next cycle. The different feedback intervals have a twofold impact on the fuzzing process. On the one hand, the fuzzing process is accelerated if the feedback interval is increased since less time-consuming monitoring steps need to be conducted. On the other hand, the level of detail a fuzzer receives is reduced if the feedback interval is increased since the number of test cases sent before receiving feedback is increased.

#### 2.4.5. RQ3—Overhead

The third research question is concerned with the overhead the models introduce. Similar to RQ1, we analyze this by comparing the model-based fuzzer to A_RANDOM and A_BASELINE. We analyze how many test cases the fuzzers are able to produce during the given time frame of 24 h and how many vulnerabilities are found during that time. Note that the models do not require a training phase before the fuzzing can start since they are trained on the fly.

We would expect that the model-based fuzzers create fewer inputs than A_RANDOM and A_BASELINE since the model queries and re-train operations take time.

#### 2.4.6. Additional Analysis

In addition to the already mentioned explicit research questions, we analyze the values the algorithms choose for their test cases and analyze the specific vulnerabilities triggered by the different fuzzers. With this, we aim to obtain more insight into the performance and the inner workings of the model-based fuzzers.

## 3. Results

This section presents the results of the experiments performed in accordance with the experimental setup presented in Section 2.4. First, we present the results on the general performance of the model-based fuzzers in comparison to the two baseline fuzzers (Section 3.1). Then, we present the results concerning the effect of the feedback dimension and the feedback intervals (Section 3.2 and Section 3.3). For the additional analysis, we present the distributions of Integer values chosen by the fuzzers in Section 3.4 and information on the throughput of the fuzzers in Section 3.5. We discuss all these results in Section 4. All the data necessary to produce the figures showed in this section as well as additional figures and data have been published (https://github.com/anneborcherding/Smarter-Evolution).

### 3.1. General Performance

First, we present the general performance of the different fuzzers in terms of the number of vulnerabilities triggered, compared to our two baselines (see Section 2.2.2 and Section 2.4.1, respectively). For these experiments, we used the default configurations, which are represented by the underlined values in Table 1. To reiterate, we use three model-based fuzzers, A_DT, A_NN, and A_SVM, which are based on a DT, NN, and SVM, respectively. For A_DT and A_NN, the model is used for selection and to guide the mutations. In contrast, the SVM is only used for the selection of test cases. As baseline, we use two different approaches: A_BASELINE and A_RANDOM. A_BASELINE corresponds to an approach that strictly selects the test cases which lead to the highest number of crashing test cases and does not change the mutation strategy. A_RANDOM uses an approach where test cases are selected at random. The mutation strategy is not changed. In total, 25 vulnerabilities are present in VulnDuT for our experiments (see Table 2).

Figure 5 and Figure 6 show the absolute number of unique vulnerabilities that have been triggered by the different fuzzers over time, and over the test cases the fuzzers chose to be sent to the SUT, respectively. Figure 5 shows the number of vulnerabilities that have been triggered by the different fuzzers over the time of the fuzzing process. Each of the shown lines represents the mean of 10 runs of a fuzzer using the respective algorithm. This graph shows that A_DT and A_NN outperform A_BASELINE and A_RANDOM. In turn, the latter two outperform A_SVM with regard to the final number of triggered vulnerabilities. However, A_SVM is able to find new bugs earlier in time.

To analyze the relative performance of the fuzzers, we conduct a statistical test on the number of triggered vulnerabilities. The results of this statistical test are presented in Table 3 and in Figure 7. Table 3 shows the pairwise *p*-values calculated using a Mann–Whitney *U* Test based on the final number of unique vulnerabilities the 10 runs of each of the fuzzers triggered. Values suggesting that two distributions of the final number of vulnerabilities are distinguishable (*p*-value <0.05) are highlighted in gray. This shows that A_DT outperforms both baseline fuzzers (A_RANDOM and A_BASELINE) significantly. Furthermore, it shows that A_SVM performs significantly worse than A_RANDOM and A_BASELINE. The final number of vulnerabilities triggered by A_DT and A_NN are not distinguishable.

In addition to the statistical analysis regarding the final value of triggered vulnerabilities, we statistically analyze the number of vulnerabilities the fuzzers trigger over time. Figure 7a,b shows the *p*-values regarding the number of triggered vulnerabilities over time, comparing the model-based fuzzers to A_RANDOM and A_BASELINE, respectively. The black horizontal line represents the significance level of 0.05. Compared to A_RANDOM, all three model-based fuzzers lead to significantly different numbers of vulnerabilities after around 20,000 s in the fuzzing campaign (Figure 7a). Note that A_SVM performs significantly differently, but triggers fewer vulnerabilities than A_RANDOM (see, for example, Figure 5). Compared to A_BASELINE, A_DT performs significantly better during the whole fuzzing campaign (Figure 7a). A_NN is only able to produce significantly better results for a few moments in time. Nevertheless, the mean number of final vulnerabilities is higher for A_NN than for A_BASELINE (see Figure 5).

Figure 6 shows the same configurations as Figure 5, but shows the triggered vulnerabilities over the number of test cases instead of time. This representation accounts for the overhead the model-based approaches introduce. As a result, Figure 6 focuses on the efficiency and effectivity of the fuzzers’ test case choices, regardless of the duration of the underlying calculations. It shows that the relative performance of the fuzzers is the same as in the previously presented analysis, which was based on the time. Furthermore, it shows that A_RANDOM is able to generate more test cases in the fixed fuzzing time of 24 h than the other fuzzers. We present details on the throughput of the different fuzzers in Section 3.5.

Next to the accumulated number of vulnerabilities, we analyze the individual vulnerabilities the fuzzers were able to trigger. Table 4 shows this data in detail. The rows of Table 4 correspond to the vulnerabilities that we implemented in VulnDuT (see Section 2.4.1). Each entry corresponds to the number of runs in which a certain vulnerability was triggered by a certain fuzzer (out of 10 runs). A_SVM, A_BASELINE, and A_RANDOM focus on the String value and find at least two linked vulnerabilities. A_NN and A_DT, however, also focus on the signed and unsigned Integer value. All the fuzzers find the first vulnerability of the three linked String vulnerabilities, except for one run of A_NN.

### 3.2. Choice of Feedback Dimension

Figure 8 shows the influence of the choice of the feedback dimension on A_DT and A_NN. We show the mean of 10 runs as well as the 95% confidence interval and restrict the plots to the results of A_DT and A_NN since all other fuzzers lead to similar results. For our experiments, we choose two configurations: (I) unidimensional feedback, where only the information whether one or more services crashed was returned, and (II) multidimensional feedback, where information on each single service was given. In the latter case, the fuzzers receive feedback with a dimension of five, one dimension for each of the services the VulnDuT provides. Figure 8 shows that the resulting number of triggered vulnerabilities does not differ significantly between fuzzers with unidimensional feedback and fuzzers with multidimensional feedback. For A_DT, A_NN, A_SVM, and A_RANDOM, the types of triggered vulnerabilities also stay the same. However, A_BASELINE shows different results if unidimensional feedback is provided. If A_BASELINE receives multidimensional feedback, the fuzzer is able to trigger all the vulnerabilities of the first set of linked String vulnerabilities (see Table 4). With unidimensional feedback, A_BASELINE is only able to trigger three of the five vulnerabilities in this set. Out of 10 runs, 10 were able to trigger the first two vulnerabilities and 3 were able to trigger the third vulnerability. This means that A_BASELINE is able to trigger more linked vulnerabilities if it receives multidimensional feedback.

### 3.3. Choice of Feedback Interval

In addition to the feedback dimension, we analyze the impact of the feedback interval (see Section 2.4.4). For a feedback interval of *x*, the fuzzer receives information on the crashed services of the SUT after every *x* test cases. Consequently, the higher the feedback interval, the lower the resolution of the feedback the fuzzer receives. Figure 9 shows the corresponding results for A_DT and A_NN. Both figures show that the choice of the feedback interval has no significant impact on the performance of the model-based fuzzers. However, similar to the impact of the feedback dimension, the feedback interval does impact the performance of A_BASELINE. This shows that the performance of the A_BASELINE fuzzer with a feedback interval of 500 triggers significantly more vulnerabilities than the performance of the A_BASELINE fuzzer with a feedback interval of 10 (*p*-value: 0.009).

### 3.4. Distribution of Integer Values

We analyze the distribution of the Integer values the fuzzers choose during the fuzzing, in order to understand how the fuzzers perform their search for vulnerabilities. Figure 10 shows the distribution of the signed Integer values chosen by the different fuzzers. We present representative runs for each of the fuzzers, while only showing one run for A_SVM, A_BASELINE, and A_RANDOM for the sake of visibility. Note that the y-axis shows the full value range of a signed Integer. High resolution figures of these results were too big to be handled by most PDF readers, but can be generated using the code that accompanies the publication of our data.

Figure 10a,b shows that both A_DT and A_NN choose the signed Integer values in a strategic way, while A_SVM, A_BASELINE, and A_RANDOM show a more random behavior (Figure 10c–e). A_DT still shows a quite high distribution of values, while A_NN focuses on one region of values. Our additional evaluation of the distributions of the unsigned Integer value shows that these distributions are similar to the distributions of the signed Integer values, which is why we do not show these results graphically. Nevertheless, the code accompanying the publication of our data includes these figures.

### 3.5. Throughput

To be able to quantify the overhead our models introduce, we measure the throughput of the fuzzers in terms of the number of test cases sent to the SUT in 24 h. Figure 11 shows the mean and standard deviation of the number of test cases sent to the SUT in 24 h, based on the data of 10 runs of each fuzzer with the default configuration. This shows that A_RANDOM is able to produce the highest number of test cases in 24 h (299,524), followed by A_BASELINE (207,882) and A_SVM (215,884), which both do not infer with the mutation. A_DT and A_NN send the smallest number of test case (79,900 and 145,372). This corresponds to a decrease in test cases sent in comparison to A_RANDOM of 27.92%, A_SVM of 51.47%, and A_NN of 73.32%, respectively. A_DT shows the smallest variation.

## 4. Discussion

After the presentation of the results of our experiments in the previous section, we discuss these results in this section and present their implications regarding the research questions formulated for this work (see Section 2.1).

### 4.1. RQ1—Model Impact

For our first research question, we analyze whether the models represent information that can be used to improve the fuzzing performance. To this end, we compare the results of the model-based fuzzer against two baselines: A_RANDOM and A_BASELINE. A_RANDOM selects test cases randomly and performs mutations at random, while A_BASELINE chooses the test cases that lead to the highest numbers of crashing services and performs random mutations on these test cases. As expected, our results presented in Section 3.1 show that A_BASELINE outperforms A_RANDOM in terms of triggered vulnerabilities (mean of 10 runs).

Furthermore, A_NN and A_DT are able to significantly outperform A_RANDOM and A_DT is able to significantly outperform A_BASELINE. These two fuzzers are based on an NN and on a DT, respectively, and both use the model to guide the mutations (see Section 2.2.3 and Section 2.2.4). Based on this observation, we conclude that the NN and DT are able to represent meaningful information, which then can be used to guide the mutations to make the fuzzing more efficient.

In contrast, A_SVM performs significantly worse than A_RANDOM and A_BASELINE in terms of triggered vulnerabilities. Note that A_SVM uses the model only to select new test cases and not to guide the mutations. Figure 5 shows that A_SVM reaches its maximum number of triggered vulnerabilities quickly and is not able to trigger further vulnerabilities over the duration of the test. A further analysis of the test case selection performed by A_SVM shows that it quickly starts to select test cases for the population that do not lead to a crash. As a result, no new vulnerabilities are found. Note that the selection of the test cases A_SVM performs leads to fewer triggered vulnerabilities than the random strategy by A_RANDOM. Future work could include an analysis of how A_SVM performs if provided with more data, along with an analysis of the resulting SVMs with a focus on how the models come to their conclusions. Furthermore, the performance of A_SVM leads to the conclusion that model-based fuzzing approaches need to ensure that the used model indeed represents meaningful information in order to avoid making the fuzzing less efficient.
**RQ1.** The fuzzer that is based on a DT (A_DT) and uses the model to guide the mutations is able to significantly outperform the two baseline fuzzers. A_NN, which is based on an NN and also uses the model to guide the mutations, significantly outperforms the random baseline. We conclude that models can help to guide the fuzzing process.


### 4.2. RQ2—Information Granularity

The second research question is concerned with the impact of the feedback dimension. We compare two configurations: (I) multidimensional feedback with full information on which services crashed and (II) unidimensional feedback with only the binary information regarding whether at least one service crashed. Our experiments show that the dimension of the feedback does not influence the performance of the fuzzer significantly (see Section 3.2). Our hypothesis was that the models would be able to improve the fuzzing if they receive more detailed information on the crashed services. However, it shows that the models lead to non-distinguishable results for both configurations in terms of the number of triggered vulnerabilities. As has been stated in Section 3, the types of triggered vulnerabilities also stay the same for A_DT, A_NN, A_SVM, and A_RANDOM, but A_BASELINE shows differences in behavior. If A_BASELINE receives multidimensional feedback, the fuzzer is able to trigger all the vulnerabilities of the first set of linked String vulnerabilities (see Table 4). With unidimensional feedback, A_BASELINE is only able to trigger fewer vulnerabilities in this set. Remember that the strategy of the A_BASELINE is to simply select the test cases that trigger the most crashes and that these linked String vulnerabilities lead to more crashes the more characters were hit (see Table 2). Based on this, it appears to be coherent that A_BASELINE is able to trigger more of these linked crashes with multidimensional feedback, since the linked vulnerabilities with several crashed services will be ranked higher than other test cases if multidimensional feedback is given. The performance of the other fuzzers does not change, which indicates that the model-based fuzzers are able to compensate for the missing information. Note that A_RANDOM performs similarly in all cases since it does not take the feedback into account at all. However, the model-based approaches find less linked String vulnerabilities in total. Nevertheless, we conclude that the models have the potential to be more resilient against missing information but would generally need better performance in finding linked vulnerabilities.

Furthermore, our experiments show that the interval of the feedback also does not influence the performance of the model-based fuzzers significantly (see Section 3.3). On the one hand, a smaller feedback interval leads to a higher resolution of the results. On the other hand, a smaller feedback interval leads to a higher overhead caused by the additional monitoring steps that need to be conducted. For the model-based fuzzers, the disadvantages and advantages seem to balance out and result in a similar performance. However, the chosen feedback interval does influence the performance of A_BASELINE as the corresponding fuzzer with a feedback interval of 500 finds significantly more vulnerabilities than the fuzzer with a feedback interval of 10. Again, the model-based approaches seem to be more resilient against changes in the level of detail of the given feedback. One possible reason for this behavior is that test cases that are sent in one interval have a higher probability of being similar to each other since they might have been generated from similar test cases. As a result, the loss of information caused by large feedback intervals may be small enough to be compensated for by the models. Future work including a more detailed analysis may provide additional insights into the underlying reasons. Furthermore, the granularity of information may have a bigger impact in real-world scenarios where the length of the test cases is higher and the number of vulnerabilities most likely smaller.
**RQ2.** The dimension and the interval of the feedback the fuzzer receives does not influence the performance of the model-based fuzzers significantly. The performance of A_BASELINE is influenced by these choices.


### 4.3. RQ3—Overhead

Our approach does not require an a priori training phase of the models used by the fuzzers since the models are trained and queried during the fuzzing process. To analyze the performance overhead the models introduce during fuzzing, we use the number of test cases the fuzzers are able to send to the SUT in 24 h as measurement (see Section 3.5). Our results show that A_RANDOM is able to produce the most test cases. This is expected since A_RANDOM does not use any model but performs the selection at random. A_BASELINE, A_NN, and A_SVM lead to comparable numbers of test case, whereas A_DT produces the fewest test cases. Nevertheless, A_DT performs significantly better regarding the number of triggered vulnerabilities than A_RANDOM and A_BASELINE in the given timeframe of 24 h (see Section 3.1). In addition, A_DT finds the vulnerabilities earlier in the fuzzing process (Figure 5). We derive that A_DT produces the fewest but the most efficient and effective test cases.
**RQ3.** Compared to the random fuzzer, the model-based approaches lead to a reduction in the number of sent test cases of 27.92% (A_SVM), 51.47% (A_NN), and 73.32% (A_DT). Nevertheless, A_DT and A_NN outperform A_RANDOM significantly, showing that they are able to produce test cases more efficiently and effectively.


### 4.4. Additional Analysis

To gain deeper insight into the workings of the different fuzzers, we analyze the Integer values chosen by the fuzzers and the vulnerabilities that the fuzzers were able to trigger during our evaluation.

The analysis of the signed and unsigned Integer values that the model-based fuzzers choose during the evaluation shows that the fuzzers follow different strategies (see Section 3.4). It shows that A_DT and A_NN select the values in a strategic way, while A_SVM, A_RANDOM, and A_BASELINE choose the values in a more random way. This corresponds to the expected behavior. Especially for A_NN, it is apparent that the fuzzer changes the values in small steps in one direction. This behavior reflects the approach of A_NN, which is to change the values in the direction of the gradient of the function the NN approximates. It is also noteworthy that each of the runs of A_NN seem to derive a different gradient, and, as a result, a different direction to change the values to (see Figure 10b). Since A_SVM does not infer with the mutation, but only influences the test case selection, it is expected that the distribution of Integer values chosen by A_SVM is similar to A_RANDOM and A_BASELINE. For future work, it would be interesting to analyze whether one could incorporate several directions of value change into one run of A_NN.

Our analysis of the vulnerabilities that have been found by the different fuzzers shows that A_SVM, A_BASELINE, and A_RANDOM focus on the String-based vulnerabilities, whereas A_DT and A_NN also focus on the Integer based vulnerabilities (see Table 4). Opposed to our expectations, A_DT was not able to find the linked vulnerabilities in the String value. It seems that the underlying DT was not able to represent the particularities of these vulnerabilities. Nevertheless, A_DT was able to find several of the Integer-based vulnerabilities. Due to the strict focus of A_BASELINE on the test cases with the highest number of crashes, it focuses highly on a small amount of test cases. As a result, A_BASELINE finds several linked vulnerabilities. In contrast, A_DT and A_NN explore the input space in a more broad way. As a result, both find several independent vulnerabilities. Interestingly, A_RANDOM does not trigger any of the Integer-based vulnerabilities. However, due to random approach, A_RANDOM will eventually find these vulnerabilities if given enough time.
**Additional Analysis.** As expected, A_NN and A_DT show a more strategic approach to choosing the signed and unsigned Integer values than A_SVM, A_BASELINE, and A_RANDOM. A_BASELINE finds more linked vulnerabilities since it focuses on already known vulnerabilities, whereas A_NN and A_DT find more independent vulnerabilities.


## 5. Future Work

Our work shows that the two model-based fuzzers, A_NN and A_DT, were able to outperform the two baseline fuzzers A_RANDOM and A_BASELINE if VulnDuT is used as a target. Future work will include evaluating the model-based approaches against real ICS and comparing their performance to state-of-the-art black box network fuzzers for ICS. With this, it will be possible to analyze how transferable the results based on VulnDuT are, and how our approach works in practice. In addition, one could analyze how it would affect the performance and throughput of the fuzzers if not the whole offspring, but only the test cases selected using the model were executed and sent to the SUT. This would reduce the number of test cases to be sent, which will save some time during the testing. However, it would also lead to less information for the fuzzer. Furthermore, a more detailed analysis of the impact of the hyperparameter choices for the different algorithms (see Section 2.3) could be conducted in the future. Another research direction would be to leverage the explainability of the underlying models of A_DT and A_NN in order to understand the underlying decision-making processes.

Our experiments show that the different feedback dimensions did not have any impact on the performance of the model-based fuzzers. One possibility to further look into this would be to add even more information to the feedback for the fuzzer to see whether this additional information improves the fuzzing performance.

Furthermore, our experiments show that the different fuzzers have different strengths in finding independent and linked vulnerabilities. An interesting approach would be to combine several algorithms in order to find a balance between finding new vulnerabilities (exploration) and finding vulnerabilities close to already known vulnerabilities (exploitation). For example, this could be achieved by combining A_DT and A_NN.

To further extend the proposed framework and approach, one could include and acknowledge state information during fuzzing. This would especially be relevant if Smevolution should be used to test stateful network protocols in the ICS domain.

## 6. Conclusions

In this work, we present Smevolution, an approach to reduce the gap between gray box testing and black box testing. During fuzzing, we train a model that approximates the inner workings of a System under Test. This model is integrated to an Evolutionary Algorithm and is used to (I) select promising test cases for the next evolutionary round and to (II) guide the mutations of the test cases. With this, we provide information to the Evolutionary Algorithm that is usually not available in a black box test setting. Our evaluation using a System under Test with artificial vulnerabilities leads to the following main insights. (I) The fuzzer based on a Decision Tree is able to significantly outperform the baseline fuzzer, which uses the Evolutionary Algorithm without the model-based support. (II) The dimensions and the intervals of the feedback the fuzzer receives and uses to train the model that we tested during our evaluation have no significant impact on the performance of the model-based fuzzers, but influence the performance of the baseline fuzzer. (III) The model-based fuzzers are able to find more independent vulnerabilities, whereas the fuzzers based only on the Evolutionary Algorithm focus on vulnerabilities close to already known vulnerabilities.

Our results can be used as a starting point to enable the application of efficient gray box test approaches to black box test settings. Especially in the domain of smart production ecosystems, efficient black box security tests are necessary due to requirements from corresponding standards and the need to test systems including parts from third-party vendors.

## Figures and Tables

**Figure 1 sensors-23-07864-f001:**
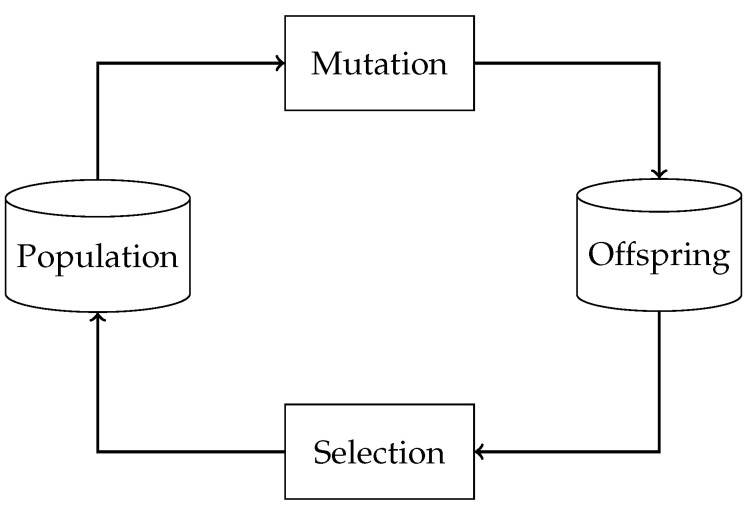
Basic version of an evolutionary algorithm. Individuals of a population are mutated to form a new offspring. From this offspring, the most promising individuals are selected based on a fitness function. These individuals form the new population.

**Figure 2 sensors-23-07864-f002:**
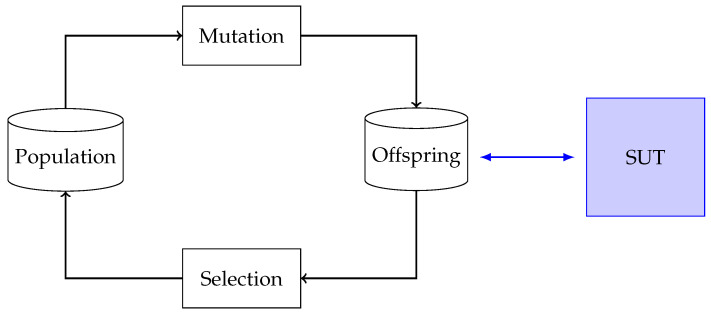
The EA Smevolution is based on. The population and offspring contain test cases for the SUT. The test cases in the population are mutated to create a new offspring. The test cases in the population are executed against the SUT. From the offspring, test cases for the population are selected based on a fitness function.

**Figure 3 sensors-23-07864-f003:**
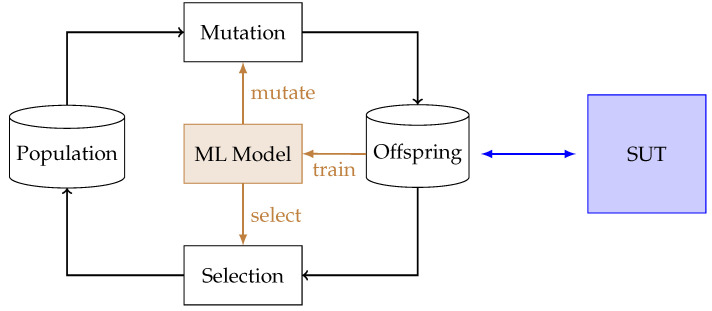
Concept of Smevolution. The model that is trained during the fuzzing process approximates a function that maps a test case to the number of services it will crash in the SUT. With this knowledge, the selection is performed, and, more importantly, the mutation is guided.

**Figure 4 sensors-23-07864-f004:**
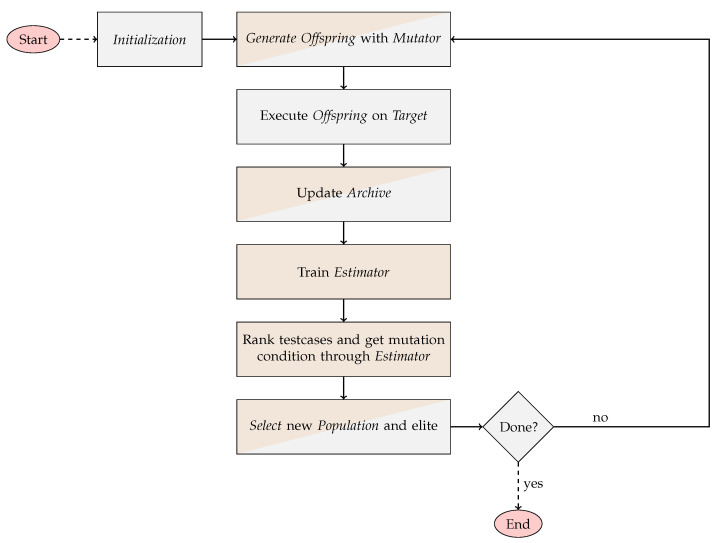
The process flow of Smevolution. Steps with a brown background are processed within the used algorithm.

**Figure 5 sensors-23-07864-f005:**
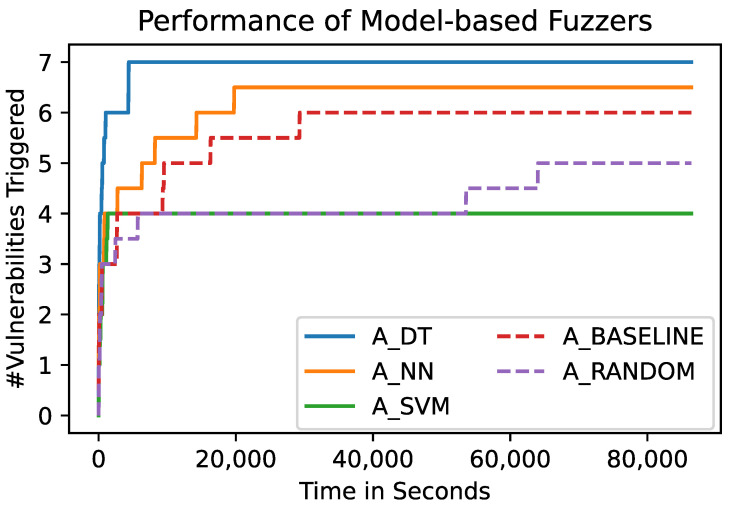
Number of unique vulnerabilities triggered by the different algorithms over time. Each line represents the mean of 10 runs of a fuzzer using the respective algorithm. A_NN and A_DT are able to outperform A_SVM, A_BASELINE, and A_RANDOM.

**Figure 6 sensors-23-07864-f006:**
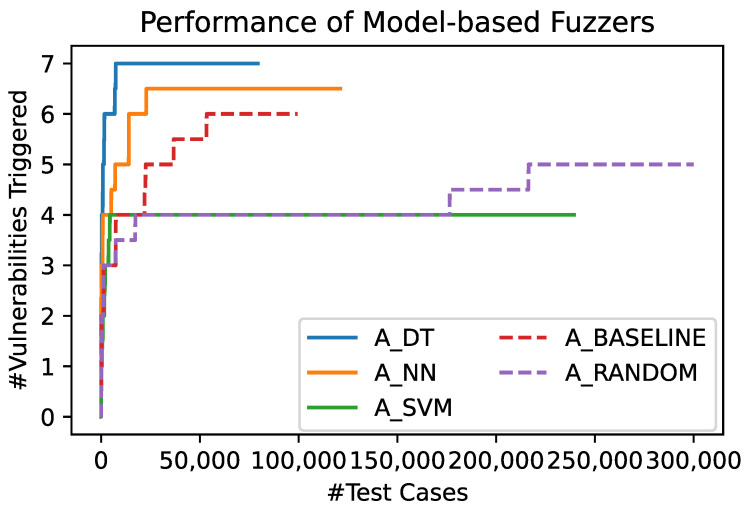
Number of unique vulnerabilities triggered by the different algorithms over the number of sent test cases. Each line represents the mean of 10 runs of a fuzzer using the respective algorithm.

**Figure 7 sensors-23-07864-f007:**
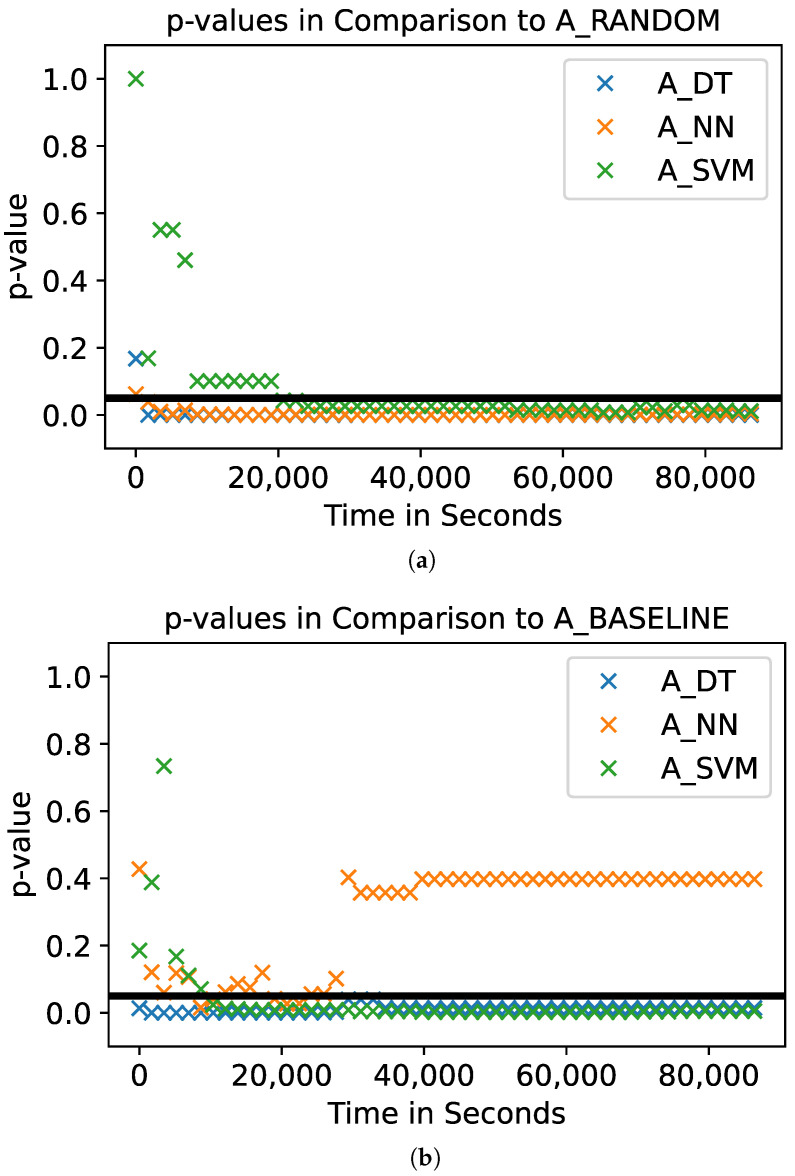
Graphical representation of the *p*-values of triggered vulnerabilities of 10 runs over the time of the fuzzing campaign (24 h). The black line represents the significance level of 0.05. A_SVM triggers significantly fewer vulnerabilities than the two baselines (see, e.g., Figure 5). (**a**) Comparison of the model-based fuzzers to A_RANDOM. The results of A_NN, A_DT, and A_SVM are significantly different from the results of A_RANDOM; (**b**) Comparison of the model-based fuzzers to A_BASELINE. The results of A_DT and A_SVM are significantly different from the results of A_BASELINE. A_NN leads to significantly different results during the fuzzing but not at the end of the 24 h.

**Figure 8 sensors-23-07864-f008:**
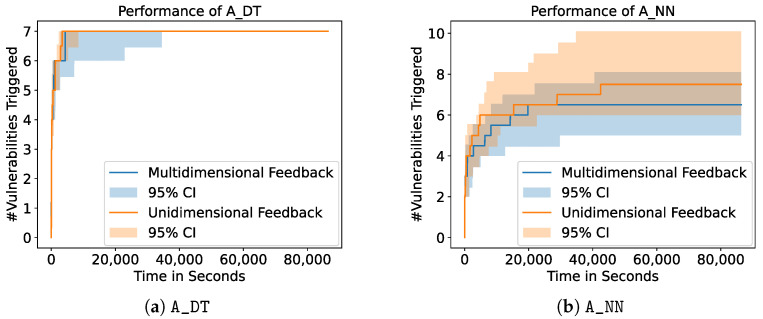
Performance of A_DT and A_NN with different feedback dimensions. The plots show the mean of 10 runs of each configuration as well as the 95% confidence interval. The different choices of feedback dimensions do not lead to significantly different results.

**Figure 9 sensors-23-07864-f009:**
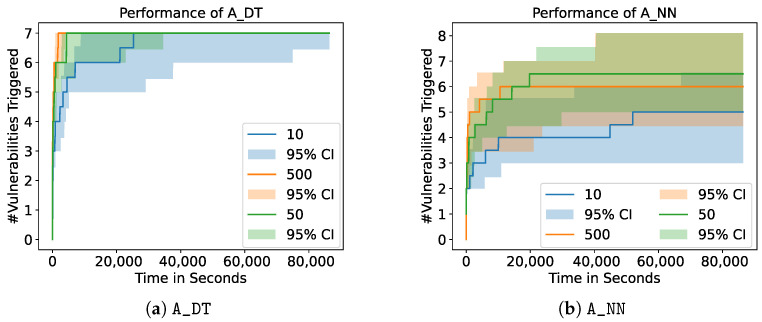
Performance of A_DT and A_NN with different feedback intervals. The plots show the mean of 10 runs of each configuration as well as the 95% confidence interval. The different choices of feedback intervals do not lead to significantly different results for the model-based fuzzers.

**Figure 10 sensors-23-07864-f010:**
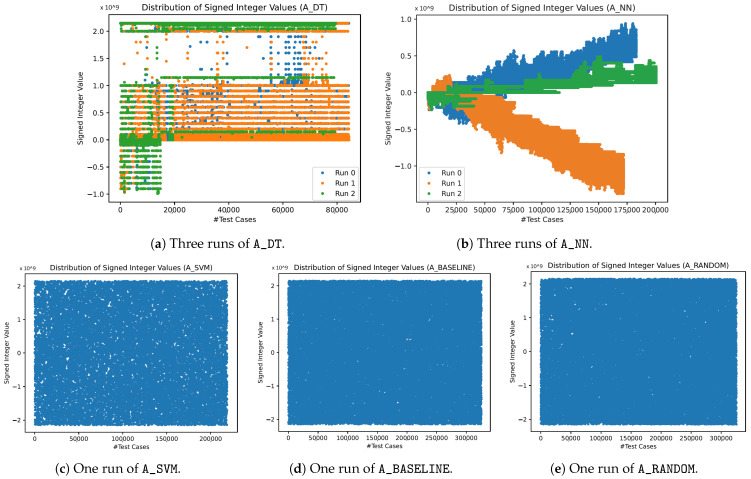
Signed Integer values that have been chosen by the different algorithms during the fuzzing. The y-axis shows the whole range for signed Integers. A_DT and A_NN choose their values strategically while the other algorithms show a more random behavior, as expected. Due to a high amount of overlapping data, we only present one run for the last three algorithms.

**Figure 11 sensors-23-07864-f011:**
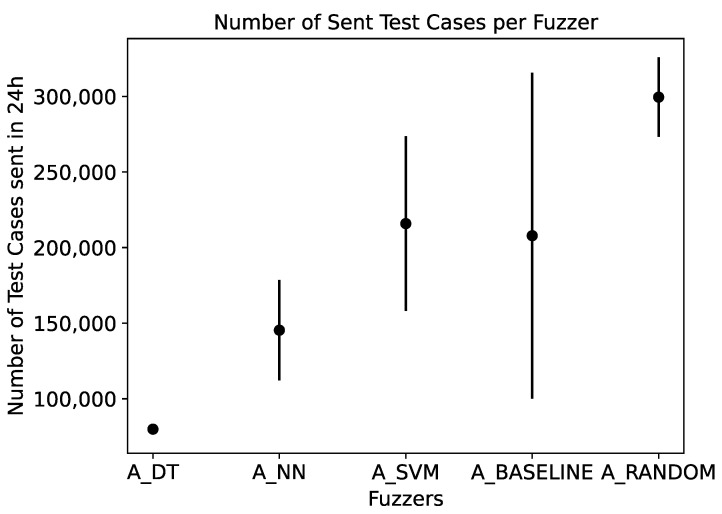
Mean and standard deviation of the number of test cases produced by 10 runs over 24 h by the different fuzzers. A_DT produces the fewest test cases.

**Table 1 sensors-23-07864-t001:** Configurations used for the evaluation based on the VulnDuT. *Algorithm* represents the algorithm used to enhance the EA, including a random algorithm and a non-model-based approach as baselines. *Feedback Dimension* corresponds to the feedback Smevolution receives from the VulnDuT, either detailed information on the crashed services or binary information as to whether at least one service crashed. With *Feedback Interval*, we define after how many test cases Smevolution will receive feedback from the VulnDuT. Accordingly, the numbers given below correspond to the number of test cases that lie between two feedback cycles. The default value is underlined.

Parameter	Values	RQ
Algorithm	A_DT, A_SVM, A_NN, A_BASELINE, A_RANDOM	RQ1
Feedback Dimension	Multidimensional, Unidimensional	RQ2
Feedback Interval	10, 50, 500	RQ2

**Table 2 sensors-23-07864-t002:** Vulnerabilities implemented in the VulnDuT for the evaluation. For each of the three fields of the communication protocol (unsigned Integer, signed Integer, and String), different vulnerabilities are implemented. This includes vulnerabilities triggered by a single value, vulnerabilities triggered by a range of values, and linked vulnerabilities. With this, we can analyze the performance of the algorithms concerning depth search and breadth search. (**a**) Unsigned Integer: Services are dropped if the unsigned Integer *x* of the test case fulfills the condition; (**b**) Signed Integer: Services are dropped if the signed Integer *y* of the test case fulfills the condition; (**c**) String: Consecutive vulnerabilities based on String matching. *X* indicates a matching Byte, dots mean the value does not matter. The String matching is performed against the String “ABCDE” and the String “FGHIJ”; (**d**) String: Consecutive vulnerabilities based on String matching. *X* indicates a matching Byte, dots mean the value does not matter. The String matching is performed against the String “KLMNOPQR”.

Condition	Dropped Services
**(a) Unsigned Integer**	
$x=2^{32}-1$	HTTP
x∈[11048576,11049576]	HTTP, ICMP
x∈[11073741824,11073742824]	HTTP, SNMP
**(b) Signed Integer**	
$y=2^{31}-1$	SNMP
$y=-2^{31}$	ICMP
y∈[1024,2024]	SNMP, ARP
y∈[−2024,−1024]	SNMP, HTTPS
**(c) String**	
X.......	ICMP
XX......	ICMP, HTTPS
XXX.....	ICMP, HTTPS, SNMP
XXXX....	ICMP, HTTPS, SNMP, HTTP
XXXXX...	ICMP, HTTPS, SNMP, HTTP, ARP
**(d) String**	
X.......	ICMP
XX......	SNMP
XXX.....	HTTP
XXXX....	ICMP
XXXXX...	SNMP
XXXXXX..	HTTP
XXXXXXX.	ICMP
XXXXXXXX	SNMP

**Table 3 sensors-23-07864-t003:** *p*-values calculated using a Mann–Whitney *U* Test based on the final number of vulnerabilities for 10 runs of the algorithms achieved after 24 h of fuzzing. Significant values (<0.05) are highlighted in gray. A_DT is able to outperform A_RANDOM and A_BASELINE significantly.

Algorithm	A_DT	A_NN	A_SVM	A_BASELINE	A_RANDOM
A_DT	-	0.068	<0.001	0.014	<0.001
A_NN	0.068	-	<0.001	0.398	0.004
A_SVM	<0.001	<0.001	-	0.005	0.011
A_BASELINE	0.014	0.398	0.005	-	0.112
A_RANDOM	<0.001	0.004	0.011	0.112	-

**Table 4 sensors-23-07864-t004:** Count of runs out of 10 that triggered a certain vulnerability of VulnDuT. For better visibility, a dash (-) represents a count of zero. See Table 2 for more information on the vulnerabilities. A_SVM, A_BASELINE, and A_RANDOM focus on the String value while A_DT and A_NN also find vulnerabilities in the signed Integer and unsigned Integer value.

Condition	A_DT	A_NN	A_SVM	A_BASE	A_RAND
$x=2^{32}-1$	10	-	-	-	-
x∈[11048576,11049576]	-	-	-	-	-
x∈[11073741824,11073742824]	-	-	-	-	-
$y=2^{31}-1$	10	-	-	-	-
$y=-2^{31}$	-	-	-	-	-
y∈[1024,2024]	10	10	-	-	-
y∈[−2024,−1024]	10	10	-	-	-
A.......	10	10	10	10	10
AB......	-	9	6	9	10
ABC.....	-	3	-	5	-
ABCD....	-	-	-	5	-
ABCDE...	-	-	-	5	-
F.......	10	10	10	10	10
FG......	-	3	2	3	5
FGH.....	-	1	-	-	-
FGHI....	-	-	-	-	-
FGHIJ...	-	-	-	-	-
K.......	10	9	10	10	10
KL......	-	-	1	1	5
KLM.....	-	-	-	-	-
KLMN....	-	-	-	-	-
KLMNO...	-	-	-	-	-
KLMNOP..	-	-	-	-	-
KLMNOPQ.	-	-	-	-	-
KLMNOPQR	-	-	-	-	-

## Data Availability

The data from our experiments as well as the Jupyter Notebook that was used to generate the figures and tables in this paper have been published (https://github.com/anneborcherding/Smarter-Evolution).

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
