# Peer review of "Smarter Evolution: Enhancing Evolutionary Black Box Fuzzing with Adaptive Models"

_sensors, 2023, doi:10.3390/s23187864_

Round 1
Reviewer 1 Report
The paper proposes the enhancement of black-box fuzzing using adaptive models. The paper is well-written and organized. However, some minor issues need to be solved:
1. Some details about the A_NN, A_DT and A_SVM architectures and parameters are needed. For example sklearn.tree.DecisionTreeClassifier() has some parameters. Do you use the implicit ones?
2. How exactly “Done?” (Figure 4) can be interpreted? Is there a metric to evaluate if the process is terminated?
3. The first table on page 10: what are the measurement units for “Feedback Intervals”? Seconds? If this time is in seconds, is it enough to retune the models or this is not necessary?
Reviewer 2 Report
· Please provide more details about the DT and NN models. What are the specific architectures and parameters used for these models?
· The authors mentioned A_RANDOM and A_BASELINE as baseline fuzzers. Could you elaborate on how these baseline fuzzers are implemented? What mutation strategies do they employ?
· The paper states that A_SVM performed worse than baseline fuzzers and other model-based fuzzers. Could you explain why the SVM model was less effective in this context? What are the limitations of using SVM for this specific task?
· In the feedback dimensions and intervals analysis, the authors found that these factors did not significantly affect model-based fuzzers. Discuss possible reasons for this lack of impact. Are there scenarios where more detailed feedback might be more beneficial?
· The paper suggests that A_DT produces the fewest test cases but performs better regarding vulnerabilities found. Please analyze deeper why A_DT achieves better efficiency and effectiveness with fewer test cases.
· The paper mentions future work evaluating the model-based fuzzers on real industrial control systems (ICS). How do you plan to address potential challenges in transferring findings from VulnDuT to real-world ICS environments with different characteristics and vulnerabilities?
· Discussing the interpretability of the models used in A_DT and A_NN would be beneficial. Being able to interpret the decision-making process of these models could provide insights into how they guide the fuzzing process.
· Since A_NN and A_DT showed complementary strengths in finding independent and linked vulnerabilities, it might be interesting to explore hybrid approaches that combine the strengths of both models. This could potentially lead to better overall vulnerability finding.
· In future work, consider investigating the sensitivity of the model-based fuzzers to different hyperparameters, such as the depth of decision trees or the architecture of neural networks. This sensitivity analysis could provide insights into fine-tuning these models for better performance.
· When extending the evaluation to real-world ICS, consider discussing potential challenges such as varying system architectures, different communication protocols, and complex network topologies. Addressing these challenges will ensure the approach's applicability in practical scenarios.
· The work involves testing vulnerabilities. It's essential to include a discussion of the ethical and security considerations. How can the approach be used responsibly to identify vulnerabilities without causing harm or exploitation?
· Consider comparing your approach with other state-of-the-art techniques for fuzz testing and vulnerability discovery in ICS to provide a more comprehensive evaluation. This could help establish the uniqueness and effectiveness of your proposed approach.
· Figures and Tables looking fine.
Minor editing of English language required
